# Mobility, Gender and Career Development in Higher Education: Results of a Multi-Country Survey of African Academic Scientists

**Heidi Prozesky** [1,*] and **Catherine Beaudry** [2,3]

1   Centre for Research on Evaluation, Science and Technology, and DST-NRF Centre of Excellence in Scientometrics and STI Policy, Stellenbosch University, Private Bag X1, Matieland 7602, Stellenbosch, Western Cape, South Africa
2   Department of Mathematics and Industrial Engineering, Polytechnique Montreal, P.O. Box 6079, Centre-ville Station, Montréal, QC H3C 3A7, Canada; catherine.beaudry@polymtl.ca
3   Centre interuniversitaire de recherche sur la science et la technologie (CIRST), Université du Québec à Montréal, C.P. 8888, succ. Centre-ville, Montréal, QC H3C 3P8, Canada
*   Correspondence: hep@sun.ac.za

**Abstract:** Empirical knowledge of the mobility of African scientists, and women scientists in particular, holds an important key to achieving future success in the science systems of the continent. In this article, we report on an analysis of a subset of data from a multi-country survey, in order to address a lack of evidence on the geographic mobility of academic scientists in Africa, and how it relates to gender and career development. First, we compared women and men from 41 African countries in terms of their educational and work-related mobility, as well as their intention to be mobile. We further investigated these gendered patterns of mobility in terms domestic responsibilities, as well as the career-related variables of research output, international collaboration, and receipt of funding. Our focus then narrowed to only those women scientists who had recently been mobile, to provide insights on the benefits mobility offered them. The results are interpreted within a theoretical framework centered on patriarchy. Our findings lead us to challenge some conventional wisdoms, as well as recommend priorities for future research aimed at understanding, both theoretically and empirically, the mobility of women in the science systems of Africa, and the role it may play in their development as academic leaders in African higher education institutions.

**Keywords:** mobility; localism; gender; career development; African scientists; higher education institutions; multi-country survey; domestic responsibilities; research output; working conditions

## 1. Introduction

The international mobility of scholars and scientists is a key feature of the global science system (Huang 2013; Knight 2008; Rostan and Höhle 2014). Mobility is largely associated with positive effects for an institution (Welch 1997) and for the mobile individuals. Mobile researchers generally have a larger international network and perform better than their non-mobile peers (Cruz-Castro and Sanz-Menéndez 2010; Franzoni et al. 2012), they publish and are cited more often (Aksnes et al. 2013; Baruffaldi and Landoni 2012; Teodorescu 2000) and have better access to funding (Cañibano et al. 2008).

Mobility is vital in an academic's career because it provides a scholar with an opportunity for interaction with other scholars and further training. The interaction facilitated through mobility, especially internationally, is important because it enables a scholar to build networks (Cruz-Castro and Sanz-Menéndez 2010), gain more skills, and also gather experience in multicultural environments outside one's comfort zone (Kerey and Naef 2004). Considering the fact that the science systems

in the majority of African countries are suffering from a continued legacy of weak institutions, the long-lasting impact of brain drain and the general lack of established support structures (Mouton 2018), the advantages of being mobile are even more pronounced for African scientists. Mobility allows them to access the top researchers in a specific field and to work in the best laboratories in the world, with state-of-the-art equipment (Beaudry et al. 2018).

It is reasonable to argue that, if women scientists are less mobile than men scientists, they may be less able than men to take advantage of these opportunities that would enhance their careers. With reference to scientists in the United States, Cole (1979) already observed in 1979 that, "[b]ecause of reduced mobility [ ... ] women may find themselves in poorer bargaining positions than men of equal talent at the same universities. Moreover, because women are less mobile, they may be less visible to other institutions. Consequently, their reduced visibility may result in fewer job offers" (84). Since then, academic work has globalized at an ever-increasing rate (Zippel 2017), and therefore women's relative lack of geographic mobility has become even more important to consider as a potential gender-related barrier in their career development into positions of academic leadership.

## 2. A Review of the Empirical Literature

Although there has not been much research on the mobility of women in science, a number of studies have noted that many women scientists are less geographically mobile than their male counterparts, in particular as a result of their greater familial responsibilities. Prozesky (2006) has reviewed this literature from 1970 to 2005, while Larivière et al. (2011), Blackmore (2014) and Zippel (2017) provided a more recent treatment of the topic. The bibliometric study by Larivière et al. (2013), using co-authorships as a proxy for extent of collaboration, found that, globally, women's publication portfolios are more 'domestic', or less international, than those of their male colleagues. Women therefore profit less from the extra citations that international collaborations accrue, a conclusion also reached by Prozesky and Boshoff (2012) with respect to South African scientists in the field of invasion ecology.

For the purpose of this article, a selection of literature on the issue of the limited mobility of African women scientists was reviewed in more depth. Tsikata (2007) noted that, because fewer African women study abroad, they usually have not had the opportunity to develop links with intellectuals and institutions in other countries. Akinsanya (2012, p. 138) argued that participation in conferences may be "out of the question if the timing is very inconvenient for the women due to 'primary responsibilities'". Similarly, Campion and Shrum (2004) found that a combination of educational and travel limitations restricts the professional networks of East African women academics, in particular those in Kenya, where men are much more likely to travel abroad for education and visit developed countries than women are. Although "abroad" is not explicitly defined by Tsikata, nor by Campion and Shrum, a closer reading of their work leads one to conclude that the term is used primarily to refer to developed countries.

In South Africa, Prozesky (2008) reported a gender-differential effect of familial responsibilities on women's early career mobility, and shows how its negative effects are amplified—for women in particular—by the lack of a dynamic research culture that characterised South African academia during apartheid. More recently, women respondents in Obers' (2015) South African case study reported that, for some of the women respondents, family responsibilities translated to limited mobility and the inability to travel to conferences. She argued that this constraint limited their access to supportive disciplinary networks which could contribute positively to their research productivity. A direct link between the limited mobility of women scholars and their lower research output in relation that of men was established by Lewison (2001) in his study of women researchers in Iceland. In South Africa, Callaghan (2016) found dependent children to be associated negatively and significantly with conference presentations, while for men, this relationship is not significant.

Our review led us to conclude that most studies tend to merely deduce, rather than directly investigate, a link between gender, mobility and career development. In addition, very few studies have

surveyed scientists across different African countries to gain insight into mobility-related challenges that confront specifically African women scientists. The contribution of this article is to address these gaps in the literature at a very broad scale, analyzing survey data collected from respondents in 41 African countries, to better understand gender differences in terms of the link between the geographic and career mobility of African scientists.

## 3. Theoretical Framework

Indications are that patriarchy still pervades the majority of African societies, and that its resulting gender-based divisions of labor both within the home and in the academic workplace have a negative impact on the careers of African women scientists (African Development Bank 2015; Akinsanya 2012; Campion and Shrum 2004; Guramatunhu-Mudiwa 2010; Olaogun et al. 2015; Tamale and Oloka-Onyango 1997; Tsikata 2007; Zewotir and Maqutu 2006). This section of our article situates this empirical literature within a theoretical framework consisting of two dimensions: the gendered division of domestic labour (as an often-cited impediment to women's mobility); and the gendered organization of paid work (in this case, academic science). Central to both these dimensions is patriarchy. While some degree of patriarchy may be universal, there is significant variation in the relative power and privilege of males and females around the world (Macionis and Plummer 2005). Our article applies to Africa where, according to Tamale and Oloka-Onyango (1997), the "forces of patriarchy [ . . . ] pervade the majority of [ . . . ] societies". At the same time, it needs to be recognised that socio-cultural differences among African countries also impact variably on the role and status of women.

The central premise of patriarchy is the systematic dominance of men over women (Giddens 2006), both in public and private spheres. In the private sphere, we find "private patriarchy": the domination of women which occurs within the household at the hands of an individual patriarch (Walby 1990). Referring to the African context, Olaogun et al. (2015, p. 302) mentioned customary practices that "hold that the man is the head of the house and has the absolute control in the decision making process of the home". Private patriarchy is an exclusionary strategy, because women are essentially prevented from taking part in public life (Walby 1990). Applied to the topic of this article, private patriarchy, and the resulting gendered division of "household production" (Macionis and Plummer 2005, p. 313), may prevent women from being geographically mobile.

Public patriarchy is more collective in form than private patriarchy (Walby 1990). Tamale and Oloka-Onyango (1997, p. 27) argued that the environment at African academic institutions is "dictated by patriarchal values and beliefs". These institutions are, therefore, what Walby (1990) referred to as patriarchal cultural institutions that prescribe acceptable standards of behavior and action. Highly educated, geographically mobile women would present, in what Wade and Ferree (2019, p. 303) term a "symbolic threat [that] potentially degrades the identity of the dominant group" (in this case, elite male global scientists), and would therefore be the object of discrimination in the form of hostile, institutionalized sexism levelled against women who do not stay at (or in the) home. In some Islamic societies in North African countries, patriarchal religious prohibitions further restrict women's movements (Macionis and Plummer 2005). Discrimination in the academic workplace can also take a more benevolent form (Wade and Ferree 2019), such as the need to protect women from the dangers of travel abroad, but thereby limiting the advantages that mobility brings.

However, the public and the private are entwined. Radical feminists such as Tamale and Oloka-Onyango (1997) argued that the roots of patriarchal oppression of women in Africa, and subsequent gender inequities in academia (the public sphere), lie in the family (the private sphere). Following Wade and Ferree (2019), women would be less mobile and, consequently, less successful at being a "global scientist" than men are, partly because of employers' beliefs about mothers and fathers. Thus, in both the public and private spheres, men may exercise their patriarchal power to align women's actions with beliefs that women should "stay at home". In the remainder of the article we present results that we interpret within this framework, but we also call upon additional theoretical

insights in instances where the results do not match the theoretical understanding of women's limited geographic mobility that we presented in this section.

## 4. Methods

Data were collected in 2016 and 2017 via a self-administered, structured web-based questionnaire, which was piloted in Zambia, and translated into French for respondents in French-speaking countries. For the purpose of this survey, researchers, scholars and scientists are defined as individuals who dedicate at least a portion of their professional activity to research. As members of a scientific community, they communicate their results and findings—primarily through peer-reviewed journal publications—to their peers. Thus, to identify and contact African scientists, we extracted corresponding authors' emails from the Web of Science and Scopus databases for each article, published from 2005 to 2015, with an institutional address in Africa. For Zambia, we also used articles in journals not indexed in the Web of Science and Scopus databases. Other sources of email addresses were the South African Knowledgebase database, the Internet, as well as snowball sampling. In total, we sent emails to more than 120,000 addresses, a little more than 20,000 were duplicates or alternative addresses that had reached the same individuals.

The survey generated data from 7513 scientists born and/or currently working in an African country. The survey questionnaire comprised sections on educational background, employment, working conditions, research output, research funding, career challenges, international mobility, collaboration, mentoring, and demographic background. For the purpose of this article, only those respondents who indicated that they were (primarily) employed in the higher education sector (71% of the sample), and held a PhD (86% of those in that sector) were included. Finally, removing the observations with incomplete questionnaires for the variables of interest for this article resulted in a sample of 3172 individuals.

The focus of this article is on the number of questions respondents were asked on mobility. We first wanted to establish the extent of their mobility. We then asked them how important they regard mobility for their own career development. In the questionnaire, mobility was defined as either working or studying "abroad". It was clearly communicated to respondents that "abroad" referred to "a country other than what they would consider their home country", which is the definition of the term we use in this article. Respondents also rated the working conditions abroad against their local working conditions. The following paragraphs first present a brief description of the sample according to the data we collected on respondents' backgrounds. We are conscious of the fact that the intersectionality between gender and race or ethnicity is an important issue to consider in a study such as this one, but for ethical and political reasons, data on race (or ethnicity) were not collected. After describing the sample, we proceed to address the various dimensions of mobility per gender, age and region. Data were analyzed with IBM SPSS Statistics 24.

## 5. Results

### 5.1. Description of the Sample

Less than a third (29%) of the sample were women, and the respondents were, on average (mean and median), 47 years of age. The youngest cohort (39 or younger) was the smallest (22%), while 42% of the respondents were 40 to 50 years of age, and the remaining 36% were older than 50. This age profile corresponds with their domestic profile: approximately half (49%) had children or other dependents aged 0–5; two-thirds (34%) had children or other dependents aged 6–18; and three-quarters (75%) had dependents aged 19 or older.

The sample included nationals of 41 different African countries. The majority of the respondents (35%) had southern African nationalities, but sizable percentages were West Africans (30%) or North Africans (26%). A relatively small percentage (8%) reported an East African nationality, while only 1% were Central African respondents (see Appendix A for a list of nationalities per region). It is relevant

for this article to note the respondents were distributed almost exactly the same in terms of the region where they were working or residing at the time of the survey (but across only 38 African countries).

On average, the respondents had received their PhDs in their mid-to-late thirties (mean = 37; median and mode = 36), but they had done so relatively recently: half (51%) graduated after 2007 (i.e., less than 10 years prior to the survey in 2016). As an indicator of field of specialization, the majority of the respondents (41%) had obtained their PhD in the natural and agricultural sciences; another 39% were divided almost equally between the social sciences (20%), and health sciences (19%); 13% had a PhD in engineering and applied technologies, while only 7% had doctoral training in the humanities.

In terms of academic rank held at the time of the survey, almost half (49%) of the sample occupied the position of professor (full or associate), 28% were senior lecturers, and 18% were lecturers. The remaining 5% were equally divided between the ranks of postdoctoral fellow and researcher/scientist. At the time of the survey, the vast majority of the respondents (91%) were employed in a permanent position, as opposed to a contract-based one.

*5.2. Gender Differences in Actual and Potential Mobility*

Campion and Shrum (2004) drew a distinction between "educational localism" and "research localism", i.e., lack of mobility in terms of (doctoral) training, and lack of work-related travel experiences in foreign countries. Applying this distinction, we first considered "educational localism", and found that by far the majority (91%) of the respondents' PhDs had been conferred by a university in a single country, most often (76%) an African country (see Table 1 below). Gender differences were small in terms of whether respondents had obtained their PhD from a single institution (90% of men and 92% of women reported this to have been the case), but the women were proportionately much more likely (84%) than the men (73%) to have graduated from an African country.

**Table 1.** Region where PhD had been obtained, by gender.

| Region | Male | Female | Total |
|---|---|---|---|
| Africa | 73% | 84% | 76% |
| Europe | 15% | 6% | 12% |
| UK | 5% | 6% | 5% |
| North America | 4% | 3% | 4% |
| Other [1] | 4% | 2% | 3% |

[1] Asia, Australasia and South America.

Mobility among young scientists in Africa has been found to be closely associated with field (Beaudry et al. 2018), and as fields differ in their gender composition, we analyzed the data further by field. Women remained less likely to have graduated abroad, except for those specializing in the humanities (82% of women, compared to 78% of men, had graduated abroad). However, it should be noted that the number of respondents in the humanities subgroup is relatively small compared to those specialized in the other fields. The largest gender difference was found in engineering and applied technologies: while 85% of women in that field had obtained their PhD in Africa, only 63% of the men had done so.

We also asked, as a measure of more recent mobility, whether respondents had studied or worked abroad over the three years preceding the survey. Although the question referred to both educational and research mobility, 76% had already completed their PhDs before 2013, and we may assume that they would have travelled for the purpose of their research work. Again, we found that only a minority (29%) of the respondents had been recently mobile. The results of a disaggregation by gender are presented in Figure 1, which shows that male respondents were proportionately more likely than female respondents to have travelled abroad.

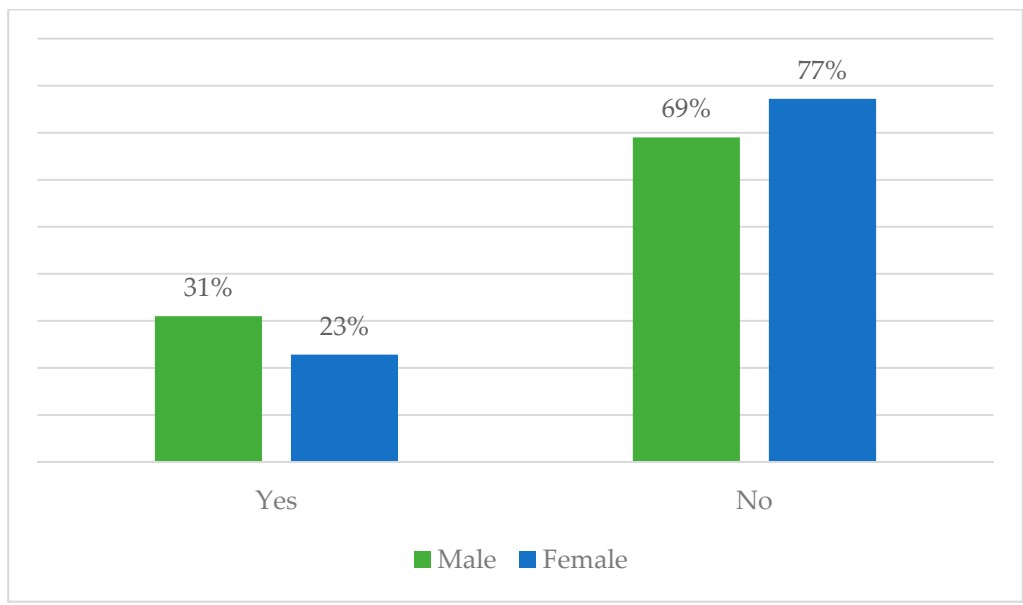

**Figure 1.** Proportions of respondents who had recently studied or worked abroad, by gender.

This pattern is the same for all the fields of specialization, although the differences between men and women was very small (0.4%) in the engineering and applied technologies, and relatively small (compared to the 8% difference for the sample as a whole) in the agricultural sciences (5%) and the natural sciences (6%). On the other hand, the gender difference was relatively large between men and women in the social sciences (10%) and health sciences (11%), and very large in the humanities (25%). Again, the results with regard to the humanities should be interpreted with caution, and this article is not primarily concerned with field differences, but it is interesting to note how the relationship between gender, mobility and field differs depending on educational localism and (primarily) research localism.

When we separated the sample according to age, males in all three age groups tended to have been more mobile, recently, than women, although the difference decreased with age, from 12% among the youngest cohort (39 or younger) to 10% among those 40–50 years old, and 7% among those older than 50. Similarly, the gender difference was observed for all ranks, but it was greatest (10%) amongst the lowest rank of lecturer, slightly smaller amongst senior lecturers (9%), and lowest amongst professors (6%).

Although less than a third of the respondents had been mobile recently, large proportions indicated that they had considered leaving the African country where they were working or residing at the time of the survey: 19% said that they had 'often' considered doing so, whilst a further 51% indicated that they had 'sometimes' thought of leaving their home country. For the remaining minority (30%), this had never been a consideration. The disaggregation by gender (Figure 2) shows that men and women differed very little in terms of whether they had often considered leaving their country, but women were proportionately more likely than men to had never considered this option.

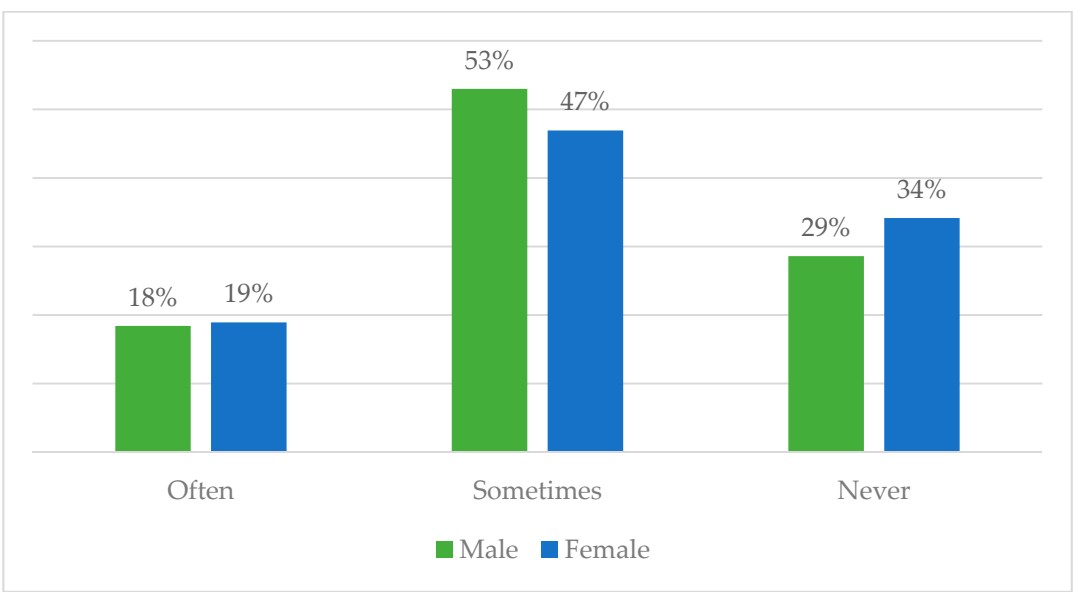

**Figure 2.** Proportions of respondents who had considered leaving the country where they work/reside, by gender.

### 5.3. International Mobility Versus Domestic Responsibilities

The explanation for the limited mobility of women that is most commonly found in the empirical and theoretical literature (as reviewed above), relates both educational and research localism of women to their family-related responsibilities, in particular their care of young children. Our survey data provided two available indicators of parenting and domestic responsibilities, but neither seems to support this argument. Compared to non-mobile women, those who had been mobile recently had, on average, a slightly higher number of children or dependents of five years or younger (1.74 compared to 1.58), and 6 to 18 years of age (1.94 compared to 1.81), which is counterintuitive. Amongst the men, we observed an opposite (but again, very small) difference between those who had been mobile and non-mobile in terms of their number of children or dependents that are five years and younger (1.85 vs. 1.91) and 6 to 18 years of age (2.27 for mobile men vs. 2.39 for non-mobile men).

The recently mobile women reported doing 60% of the care and general housework themselves, rather than delegating it to a partner or someone else. Amongst their non-mobile counterparts, the corresponding percentage is slightly lower (58%), which is again counterintuitive. The recently mobile men, on the other hand, reported doing 35% of the care and general housework themselves, rather than delegating it to a partner or someone else. Amongst their non-mobile counterparts, the corresponding percentage is only one percentage point higher (36%).

### 5.4. Gender Differences in Perceived Impact of Mobility on an Academic Career

Moving on from the possible causes to the impact of localism, we analyzed how African scientists perceived the impact of a lack of mobility on academic careers, and whether gender differences existed in this regard. All respondents, whether mobile or not, were asked to indicate to what extent ('not at all'; 'to some extent'; or 'to a large extent') a lack of mobility opportunities may have impacted negatively on their careers as academics. The variable was recoded into a binary form ("No" and "Yes", with the latter including "To some extent" and "To a large extent") for ease of comparison.

Respondents of both genders indicated that a lack of mobility opportunities had impacted—at least to some extent—negatively on their careers. However, an interesting finding is that women were proportionately less likely than men to report such a negative effect Figure 3.

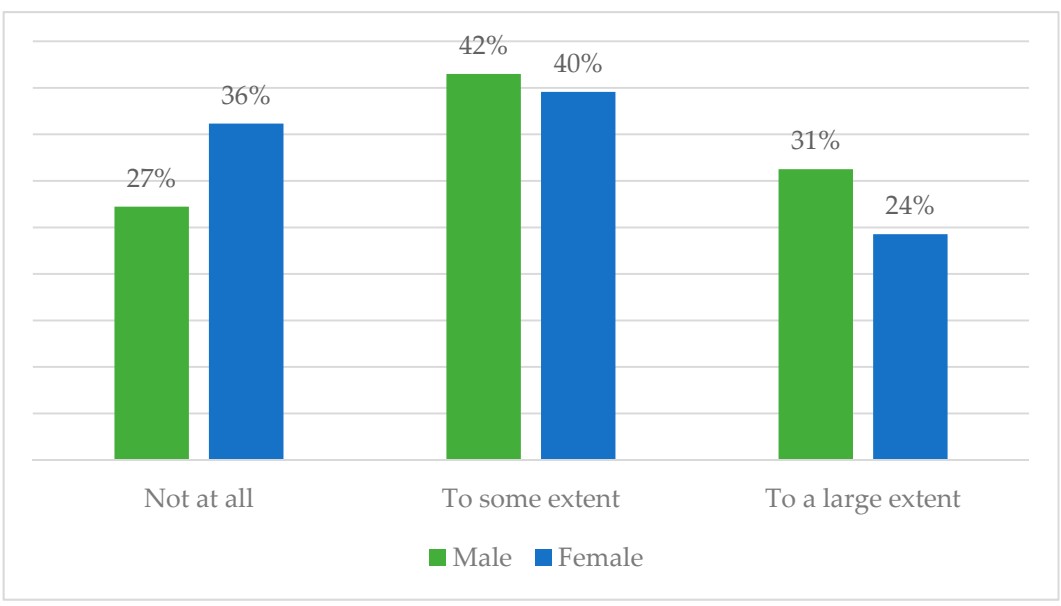

**Figure 3.** Perceived impact of lack of mobility opportunities, by gender.

If we narrow our focus to only those respondents who had worked or studied abroad, the majority indicated that this mobility had been "essential" (39%) or "very important" (45%) for their career development (only 4% rated it as less than "important").

Gender differences in this regard were only found in the two most extreme, positive response categories (Figure 4) and women were proportionately more likely than men to rate having studied/worked abroad as "essential" (the most extreme response category) for their career development.

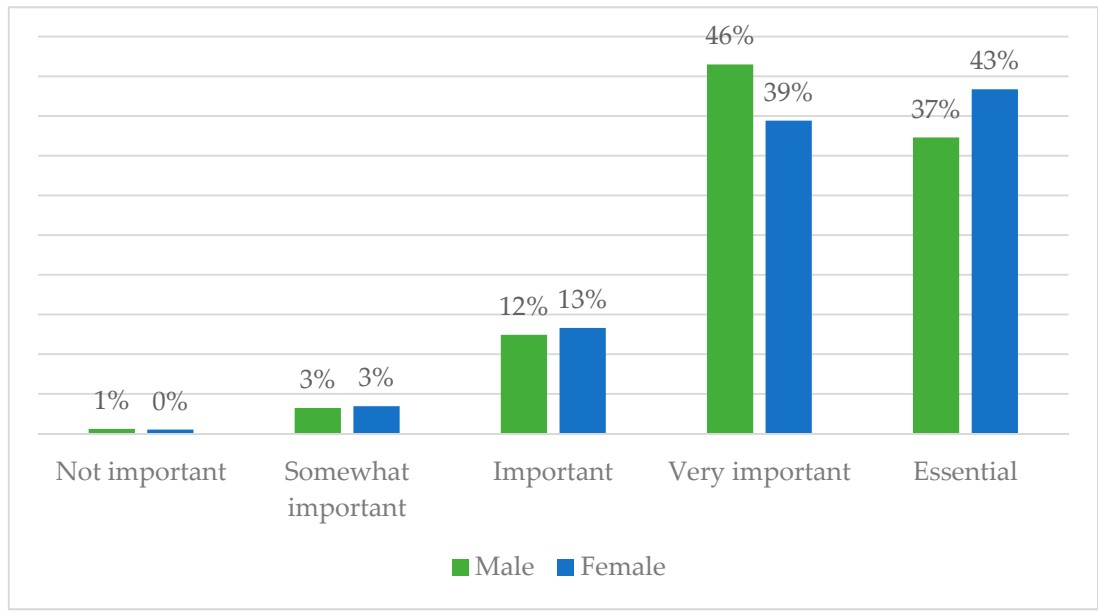

**Figure 4.** Rating of the importance of having studied/worked abroad for career development, by gender.

It is interesting that when we consider all respondents (mobile and non-mobile) women, were less likely than men to report a negative effect of a lack of mobility. But when we consider only the mobile respondents, women were more likely than men to consider that mobility essential for their career

development. This may indicate that women only realise the importance of mobility for their careers until they have actually been mobile.

### 5.5. Women Academics' Comparison of Study/Working Conditions Abroad to Those in Their Home Country

One way in which one may understand why mobile women consider mobility experiences so important to their careers, is to investigate the recently mobile women respondents' comparison of the study or working conditions in their home country with the ones they had experienced abroad. In the questionnaire, respondents were provided with six aspects on which to draw comparisons: (1) employment or job security; (2) work–family balance; (3) training opportunities; (4) opportunities for research collaboration; (5) research resources; and (6) research funding opportunities. Response options were again recoded, from the original five in the questionnaire ("Much worse abroad"; "Somewhat worse abroad"; "About the same"; Somewhat better abroad"; and "Much better abroad") to the following three categories: "Somewhat or much worse"; "About the same"; and "Somewhat or much better". We focus on the last category in the presentation of our results in Figure 5 below.

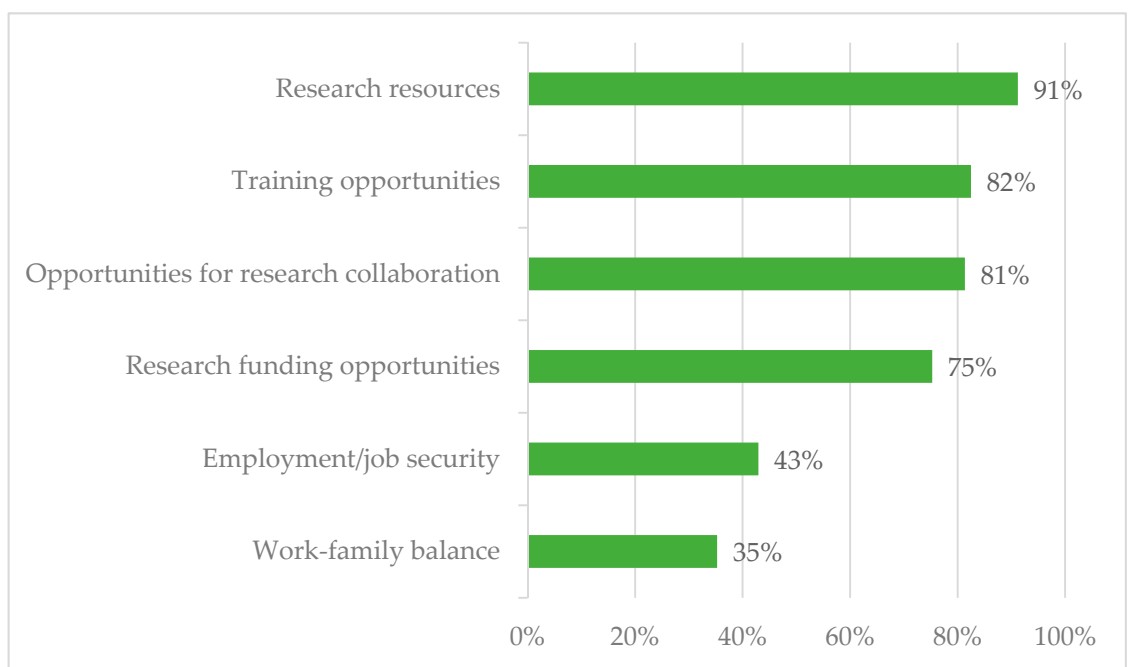

**Figure 5.** Women's rating of study/working conditions abroad compared to those in their home country.

By far the largest percentage of recently mobile women (91%) rated the country abroad as (somewhat or much) better in terms of research resources (personnel, scientific literature, material, etc.). Relatively large percentages of women also rated opportunities for training (82%), research collaboration (81%) and research funding (75%) of the country abroad as superior. On the other hand, less than half of the respondents rate the country abroad as better in terms of employment/job security (43%) and, notably, work–family balance (35%).

### 5.6. The Relationship between Mobility, Gender, and Academic-Career-Related Variables

In addition to measuring women academics' perceptions of the implications of mobility (or lack thereof) for their careers, we investigated the relationship between gender, recent mobility and three career-related variables: research output, international collaboration, and receipt of research funding (recognising that these variables may be both antecedents and results of their mobility). We found that, in comparison with their non-mobile counterparts, women who had recently travelled abroad produced, on average, more research articles, but the difference is very small (7.66 self-reported articles

in peer-reviewed journals, compared to 6.83). For men, the results are somewhat counterintuitive, as non-mobile men produced, on average, more research articles than their non-mobile counterparts. Although the difference is again very small (9.15 compared to 8.12), our results seem to corroborate those presented earlier, in Figure 4, i.e., that women were proportionately more likely to perceive mobility as essential for their career development. Mobile women were also proportionately much more likely to collaborate internationally (47% of them, compared to the 35% of non-mobile women, collaborated often or very often with researchers at institutions outside Africa), and to have been primary recipients of research funding (54% vs. 45%).

## 6. Discussion and Conclusions

The literature leads one to conclude that less mobile scientists are bereft of opportunities that would enhance their careers. Mobility seems to be gendered, however: a number of relatively small-scale studies have noted that many women scientists are less geographically mobile than their male counterparts. It is therefore important to investigate, in more depth and on a greater scale, the mobility of scientists from a gender perspective. Our large-scale survey, the first of its kind in Africa, has shown how African scientists, both male and female, value being able to study and work abroad, but women are proportionately more likely than men to perceive mobility as essential for their career development.

Although some of the African scientists we surveyed have had the opportunity to study or work abroad, it is still noteworthy that the majority, and especially young women in the lower academic ranks, have been less mobile than males in the same youngest age group and lower ranks. The gender difference in mobility decreased as age (and rank) increased, which seems to indicate that, as women advance in chronological age and in their careers, family-related duties become less of an impediment to being mobile.

The minority of women who have benefitted from international visits, reported the advantages to them and their careers in terms of research resources, but not necessarily work–life balance. We also found a clear link between mobility on the one hand, and research output, collaborating internationally, and securing funding, on the other. Reduced mobility therefore has the potential to negatively impact on the career development of women scientists, including their moving in to academic leadership positions.

What are the 'glass fences' (Zippel 2017), i.e., the various gendered challenges, that restrict women's mobility? The body of empirical literature we reviewed supports the notion that the greater family demands women scientists need to attend to play a central role, especially in Africa. In most African societies, and in both the public and private spheres, men tend to exercise their patriarchal power to align women's actions with beliefs that women should "stay at home".

However, our data on parenting and domestic responsibilities do not seem to support this argument. Compared to non-mobile women, those who have been mobile recently have, on average, a slightly higher number of children or dependents 18 or younger, and report doing a slightly greater percentage of housework themselves (rather than delegating it to a partner or someone else). In the case of both these measures, however, the difference between mobile and non-mobile women is very small. Similarly, for men, we also observe a a very small difference between mobile and non-mobile men in terms of both their number of children and the proportion of the household chores that they perform themselves.

Our results therefore challenge the conventional wisdom that family barriers are a significant obstacle to women academics' mobility. Other gender-related barriers to mobility, which are linked to patriarchal customs, may be relevant and should be investigated, especially in the African context. For example, in some African countries restrictions on women's interactions with non-familial men and norms inhibit their movement outside of the local area (Miller and Shrum 2012; Tamale and Oloka-Onyango 1997). Additionally, when institutions (public patriarchy) and individuals (private

patriarchy) construct safety abroad as a gendered issue (Zippel 2017), benevolent sexism may prevent women from travelling abroad on the grounds that it is unsafe.

In general, we may conclude that less mobile women do not comfortably fit, in Zippel's (2017) terms, the ideal of an elite male global scientist with the personal, social, and academic resources to climb fences, but they aspire to that ideal. Career-support programmes should therefore continue to target gender inequalities in terms of these resources, thereby fostering international collaboration for female researchers. However, we also find that men are proportionately more likely than women to report the negative effect of a lack of mobility on their careers, and there is some indication that women only realise the importance of mobility for their careers until they have actually been mobile. This lead us to suggest, for further research and interventions, addressing women's own career expectations and empowering non-mobile women with information on the negative effect that a lack of mobility may have on their careers.

To understand gender differences in career-related expectations, Stouffer et al.'s (1949) concept of relative deprivation may be useful (*cf.* Prozesky and Mouton 2019). Relative deprivation has been used to explain the paradox that women report higher levels of job satisfaction than do men, although, by most objective standards, women's jobs are worse than men's (Clark 1997). Similarly, we would argue that, because of normative or other restrictions of women's geographic mobility, their expectations of mobility are lower than men's. This corresponds with our finding that women are proportionately more likely than men to have never considered leaving the African country where they were working or residing at the time of the survey.

Applying a broader lens, gender differences in career expectations and resulting career-advancing behavior, may also be understood in terms of Cole and Fiorentine's (1991) theory of "normative alternatives". According to these authors, "[w]hereas occupational success is virtually the only route to adult status open to men, women can attain adult status in an affiliative way through marriage and family. Because there are normative alternatives open to women, which are not open to men, there is substantially more pressure on men to be occupationally successful" (222).

However, as our theoretical framework suggests, the role of patriarchy should also be taken into account. Men's lack of normative alternatives reminds us of the argument that "men's self-esteem comes, in part, from being a man doing men's work", which leads them to discriminate against women in masculinized occupations who present a symbolic threat (Wade and Ferree 2019, p. 304). In many African countries (the focus of our article), patriarchy prescribes a lower level of occupational attainment for women than for men. In such socio-cultural contexts, women scientists with a PhD and a permanent position in higher education (the survey respondents analyzed for this article), present a symbolic threat to the patriarchal powers that be. If women have internalized the patriarchal notion of "knowing their place" at home and/or at the margins of science (Bevan and Gatrell 2017), they should, therefore, experience less cultural pressure than men to concentrate all their energies on aspiring to the ideal of being a global scientist.

Such theoretical insights on the maintenance of sociological constructions of gender by both women and men, also remind us of the inherent bias in most research (including our study) on women aspiring the academic leadership and the barriers they face. As argued by Prozesky (2018), the data on these issues are collected almost exclusively from the "surviving superwomen", the ones who have successfully negotiated the social constructions of gender that limit mobility. A bias in favour of these women, from whom data can relatively easily be collected, unfortunately leads to an underestimation of the nature and extent of the barriers faced by women who exist in the margins of globalized academia, but definitely deserve to be heard.

**Author Contributions:** The study which produced the data analysed for this article was primarily conceptualised and designed by C.B., with H.P. providing some input during proposal writing. H.P. adapted, for the African context, the data collection instrument previously designed by C.B. Data analysis and interpretation was conducted by H.P., as was original draft preparation. C.B. provided revision of the draft, and conducted additional analysis. Project leadership, management and administration, as well as funding acquisition, were the responsibilities of C.B.

**Funding:** This research was funded by the IDRC (Canada), grant number 107987-001; the Robert Bosch Stiftung (grant number 11.5F081.0006.0), and the DST-NRF Centre of Excellence in Scientometrics and Science, Technology and Innovation Policy (SciSTIP), National Research Foundation-grant number 91488.

**Conflicts of Interest:** The authors declare no conflict of interest. The founding sponsors had no role in the design of the study; in the collection, analyses, or interpretation of data; in the writing of the manuscript, or in the decision to publish the results.

## Appendix A. Respondents' Nationalities by Region

| | |
|---|---|
| Central African | Central African Republic; Congo-DRC; Congo-Republic; Gabon |
| East African | Burundi; Comoros; Ethiopia; Kenya; Madagascar; Mauritius; Somalia; Sudan; Tanzania; Uganda |
| North African | Algeria; Egypt; Morocco; Tunisia |
| Southern African | Angola; Botswana; Lesotho; Malawi; Mozambique; Namibia; South Africa; Swaziland; Zambia; Zimbabwe |
| West African | Benin; Burkina Faso; Cameroon; Chad; Cote d'Ivoire; Ghana; Guinea; Mali; Mauritania; Niger; Nigeria; Senegal; Sierra Leone; Togo |

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
