# Peer review of "Mobility, Gender and Career Development in Higher Education: Results of a Multi-Country Survey of African Academic Scientists"

_socsci, doi:10.3390/socsci8060188_

Round 1

Reviewer 1 Report

Overall the paper is an interesting piece. The impact of mobility on academic progression is a very important issue, so congrats to the authors for tackling it.

There were a few areas that needs addressing

What is the theoretical framework of the paper? At some places the paper read as though it was challenging Zippel's views on gendered mobility of women in academia, but fell short of clarity. The topic is quite significant so must be situated within a clearly defined theoretical framework. I would suggest that the authors flesh out "patriarchy" more clearly as a theoretical option or "gendered world of work" as discussed by Wade, L., & Ferree, M.M. (2019). Gender: ideas, interactions and institutions. (Second edition). New York: W.W. Norton & Company Inc.

The methods section also needs improving. It is important to situate the whole study in a methodological paradigm. Was it interpretive, pragmatist, realist or critical. In line 109 the authors mentioned that they conducted interviews. Interviews with who, where and how many? Was the research a mixed method study? If yes, which type was applied 1) sequential explanatory; 2) sequential exploratory; 3) sequential transformative; 4) concurrent triangulation; 5) concurrent nested; and 6) concurrent transformative (see Creswell 2003).

Line 92 states "7500 scientists born and currently working in an African country".  Did this include white Africans in south Africa, Zimbabwe on ethnicity was not controlled for? If Yes authors need to be clear on this. g 122 states southern African nationalities". Because clearly the experiences of women and men based on race would be different (intersectionality).

Authors mentioned in several places eg (lines 151, 177) that they controlled for some variables, but dont tell us how they did that.

Typographical errors in Line 150 (scientist), Line  191 "who have considered leaving"

What is the definition of abroad as used in the paper? It would be unfortunately neocoloniastic or westerncentric to argue that if a Nigerian academic studied in Zimbabwe, it is not a study abroad. Mobility is travel regardless of the study so I would suggest reconceptualisation of the term "abroad".

The findings would be best presented in the past tense. The present tense makes it difficult to follow. For example the "is" in 201 should be "was"

Author Response

Overall the paper is an interesting piece. The impact of mobility on academic progression is a very important issue, so congrats to the authors for tackling it.

There were a few areas that needs addressing

What is the theoretical framework of the paper? At some places the paper read as though it was challenging Zippel's views on gendered mobility of women in academia, but fell short of clarity. The topic is quite significant so must be situated within a clearly defined theoretical framework. I would suggest that the authors flesh out "patriarchy" more clearly as a theoretical option or "gendered world of work" as discussed by Wade, L., & Ferree, M.M. (2019). Gender: ideas, interactions and institutions. (Second edition). New York: W.W. Norton & Company Inc.

Response: Thank you for pointing out the need for a theoretical framework. We added a new section detailing our theoretical framework, in which we flesh out patriarchy in more detail, by including the theoretical insights of, primarily, Walby (1990) and (as suggested) Wade and Ferree (2019). We also integrated those insights into the last section (6, Discussion and conclusion), which we believe has increased the level of sophistication of the interpretation of our results.

The methods section also needs improving. It is important to situate the whole study in a methodological paradigm. Was it interpretive, pragmatist, realist or critical. In line 109 the authors mentioned that they conducted interviews. Interviews with who, where and how many? Was the research a mixed method study? If yes, which type was applied 1) sequential explanatory; 2) sequential exploratory; 3) sequential transformative; 4) concurrent triangulation; 5) concurrent nested; and 6) concurrent transformative (see Creswell 2003).

Response: Some interview data, collected for the purpose of a larger study, were initially considered for inclusion in the manuscript, but those data did not add much value to the manuscript, which was already quite lengthy and complex. The reference to interviews should not have been included, and has been deleted. This should address the issues raised, including the methodological paradigm. It is quite clear that the manuscript reports on quantitative survey data which is, per definition, realist. However, we could state that, if Reviewer 1 deems it necessary, even after taking into account that the manuscript does not report on a mixed methods study. We would like to note that, in consideration of Reviewer 1’s comments, we did make some changes to the methods section by adding detail and, hopefully, more clarity.   

Line 92 states "7500 scientists born and currently working in an African country".  Did this include white Africans in south Africa, Zimbabwe on ethnicity was not controlled for? If Yes authors need to be clear on this. g 122 states southern African nationalities". Because clearly the experiences of women and men based on race would be different (intersectionality).

Response: Yes, the sample includes white South Africans and Zimbabweans. Although we are aware of the importance of intersectionality, the project team decided not to include a question on race or ethnicity in the questionnaire. There were two reasons for this. First, such a question would have been difficult to justify to the relevant ethics boards. Questions about ethnic background were not well received at that time in Canada (the funding for the project was sourced, primarily, from the International Development and Research Council which is located in Canada), and research that classified survey respondents by “race” (white vs. black) is a highly sensitive political issue in post-Apartheid South Africa (the survey was sent out from a South African higher education institution, Stellenbosch University; see, for example, https://www.sun.ac.za/english/Lists/news/DispForm.aspx?ID=6426 ). Classifying African respondents from outside South Africa (or perhaps, Zimbabwe) would lead to a reification of a social construct. However, we are conscious that being unable to take into account intersectionality may a limitation of our research and mention so in the manuscript (at the end of the Methods section).

Authors mentioned in several places eg (lines 151, 177) that they controlled for some variables, but dont tell us how they did that. 

Response: This is an error of English, which we apologise for. We separated the sample according to this variable, but did not control for the variable as if we were doing inference analysis. This was corrected in the paper.

Typographical errors in Line 150 (scientist), Line  191 "who have considered leaving"

Response: Thank you for the close reading. These errors were corrected in the paper.

What is the definition of abroad as used in the paper? It would be unfortunately neocoloniastic or westerncentric to argue that if a Nigerian academic studied in Zimbabwe, it is not a study abroad. Mobility is travel regardless of the study so I would suggest reconceptualisation of the term "abroad".

Response: We appreciate that Reviewer 1 identified this potential misunderstanding. To address the issue, we included the following clarification in our Methods section: “It was clearly communicated to respondents that ‘abroad’ referred to ‘a country other than what they would consider their home country’, which is the definition of the term we use in this article”. We also revisited the literature we reviewed to clarify the use of the term in said literature (see section 2).

The findings would be best presented in the past tense. The present tense makes it difficult to follow. For example the "is" in 201 should be "was"

Response: We have made the necessary corrections to use the past tense.

Reviewer 2 Report

This is an important article. It would be nice however to know more about the background of people who made up the sample. Which women and men went overseas? Is there a correlation with socio-economic or chiefly tribal background? Has that migration furthered their career in Africa?

Are there strong connections between colonial history and migration pathways? Are people migrating inside historical colonial realms inside Africa as well as outside Africa?

Are different gender roles and expectations different in different nations? As the article is reasonably short, maybe some of these questions could be addressed?

Author Response

This is an important article. It would be nice however to know more about the background of people who made up the sample. Which women and men went overseas? Is there a correlation with socio-economic or chiefly tribal background?

Response: These are important questions, but unfortunately we cannot answer them, as we did not collect data on socio-economic or tribal background.

Has that migration furthered their career in Africa?

Response: We collected data on respondents’ in perceived impact of mobility on their academic careers (see section 5.4). Unfortunately it was not possible to collect detailed data on actual career trajectories an objective measure of career success.

Are there strong connections between colonial history and migration pathways? Are people migrating inside historical colonial realms inside Africa as well as outside Africa?

Response: Again, these are interesting questions, and we appreciate that Reviewer 2 raises them. They are, however, beyond the scope of this manuscript and should receive due attention in a separate paper.

Are different gender roles and expectations different in different nations? As the article is reasonably short, maybe some of these questions could be addressed?

Response: We have not been able to find any relevant literature on this issue, and we did not collect data that directly measure gender roles and expectations. However, it is reasonable to assume that (African) nations are not homogenous in this regard. We added (to a new section 3, Theoretical framework) the following: “While some degree of patriarchy may be universal, there is significant variation in the relative power and privilege of males and females around the world (Macionis and Plummer, 2005). Our article applies to Africa where, according to Tamale and Oloka-Onyango (1996), the ‘forces of patriarchy […] pervade the majority of […] societies’”. Variations within African countries was beyond the scope of this paper, but we recognise this variation in the sentence that now follows the previous one in the manuscript.